# Therapeutic Anti-Tumor Efficacy of DC-Based Vaccines Targeting TME-Associated Antigens Is Improved When Combined with a Chemokine-Modulating Regimen and/or Anti-PD-L1

**DOI:** 10.3390/vaccines12070777

**Published:** 2024-07-15

**Authors:** Jennifer L. Taylor, Kathleen M. Kokolus, Per H. Basse, Jessica N. Filderman, Chloe E. Cosgrove, Simon C. Watkins, Andrea Gambotto, Devin B. Lowe, Robert P. Edwards, Pawel Kalinski, Walter J. Storkus

**Affiliations:** 1Departments of Dermatology, University of Pittsburgh School of Medicine, Pittsburgh, PA 15213, USA; taylorj2@upmc.edu (J.L.T.); cec126@pitt.edu (C.E.C.); 2Departments of Immunology, Roswell Park Comprehensive Cancer Center, Buffalo, NY 14203, USA; kathleen.kokolus@roswellpark.org (K.M.K.); per.basse@roswellpark.org (P.H.B.); pawel.kalinski@roswellpark.org (P.K.); 3Departments of Immunology, University of Pittsburgh School of Medicine, Pittsburgh, PA 15213, USA; jef116@pitt.edu; 4Departments of Cell Biology, University of Pittsburgh School of Medicine, Pittsburgh, PA 15213, USA; swatkins@pitt.edu; 5Departments of Surgery, University of Pittsburgh School of Medicine, Pittsburgh, PA 15213, USA; gambottoa@upmc.edu; 6Department of Immunotherapeutics and Biotechnology, Jerry H. Hodge School of Pharmacy, Texas Tech University Health Sciences Center, Abilene, TX 79601, USA; devin.lowe@ttuhsc.edu; 7Departments of Obstetrics, Gynecology and Reproductive Sciences, University of Pittsburgh School of Medicine, Pittsburgh, PA 15213, USA; edwarp@upmc.edu; 8UPMC Hillman Cancer Center, Pittsburgh, PA 15213, USA; 9Departments of Pathology, University of Pittsburgh School of Medicine, Pittsburgh, PA 15213, USA; 10Departments of Bioengineering, University of Pittsburgh School of Medicine, Pittsburgh, PA 15213, USA; 11W1151 Thomas E. Starzl Biomedical Sciences Tower, 200 Lothrop Street, Pittsburgh, PA 15213, USA

**Keywords:** anti-PD-L1, checkpoint inhibitors, chemokines, dendritic cells, immunotherapy, inflammation, melanoma, tumor-infiltrating lymphocytes, vaccines, vascular normalization

## Abstract

We previously reported that dendritic cell (DC)-based vaccines targeting antigens expressed by tumor-associated vascular endothelial cells (VECs) and pericytes effectively control tumor growth in translational mouse tumor models. In the current report, we examined whether the therapeutic benefits of such tumor blood vessel antigen (TBVA)-targeted vaccines could be improved by the cotargeting of tumor antigens in the s.c. B16 melanoma model. We also evaluated whether combination vaccines incorporating anti-PD-L1 checkpoint blockade and/or a chemokine-modulating (CKM; IFNα + TLR3-L [rintatolimod] + Celecoxib) regimen would improve T cell infiltration/functionality in tumors yielding enhanced treatment benefits. We report that DC–peptide or DC–tumor lysate vaccines coordinately targeting melanoma antigens and TBVAs were effective in slowing B16 growth in vivo and extending survival, with superior outcomes observed for DC–peptide-based vaccines. Peptide-based vaccines that selectively target either melanoma antigens or TBVAs elicited a CD8^+^ T cell repertoire recognizing both tumor cells and tumor-associated VECs and pericytes in vitro, consistent with a treatment-induced epitope spreading mechanism. Notably, combination vaccines including anti-PD-L1 + CKM yielded superior therapeutic effects on tumor growth and animal survival, in association with the potentiation of polyfunctional CD8^+^ T cell reactivity against both tumor cells and tumor-associated vascular cells and a pro-inflammatory TME.

## 1. Introduction

Therapeutic cancer vaccines have been studied in various forms and with differing degrees of operational understanding for more than 100 years but have undergone a resurgence in research interest and clinical translation over the past decade [1,2,3]. Effective cancer vaccines target host dendritic cells (DCs) in vivo, allowing for consequent (cross-)priming of antigen-specific T cells [4]. Alternatively, ex vivo-generated DCs directly loaded with antigens in treatment formulations are used to reduce the requirement for antigen delivery to host APCs within sites of vaccination [5]. Expectedly, antigens/peptides employed in cancer vaccines have derived primarily from intrinsic tumor cell-associated antigens (including tumor neoantigens) [6,7]. However, the tumor microenvironment also contains a myriad of stromal cell populations exhibiting altered phenotypes compared to their counterparts in normal tissues, allowing for their selective targeting by antigen-specific T cells for potential therapeutic benefit [8]. Relevant to this latter point, we and others have defined non-mutated antigens that are differentially expressed by tumor-associated vs. normal tissue-associated pericytes and vascular endothelial cells (VECs), with targeted gene- or peptide-based vaccination of tumor-bearing mice resulting in T cell-mediated control of tumor growth/progression [9,10,11,12]. We also subsequently demonstrated that DC–peptide-based vaccines targeting six tumor blood vessel antigens (TBVAs) were safe and effective in promoting immunologic and/or objective clinical responses in 46% of treated advanced-stage melanoma patients [13].

Despite the safety and translational/clinical promise of DC-based vaccines, there remains substantial room for improvement in treatment design since the recruitment and maintenance of vaccine-induced T cell populations into/within tumors in vivo remains a limitation for optimal therapeutic benefit achieved by vaccination against tumor antigens [14,15]. In an attempt to develop vaccine approaches exhibiting improved anti-tumor efficacy in the subcutaneous (s.c.) B16 melanoma model, we examined combination vaccine regimens that coordinately target both intrinsic melanoma antigens and TBVAs and/or which contain treatment agents known to enhance the recruitment of T cells into tumors (i.e., chemokine-modulating regimen, CKM) [16] or to prevent exhaustion and sustain/recover the polyfunctionality of T cells (i.e., anti-PD-L1) [17,18].

We report that DC-based vaccines coordinately targeting both melanoma antigens and TBVAs outperformed vaccines targeting either melanoma antigens or TBVAs alone when considering tumor growth control or overall survival of tumor-bearing mice as endpoints. Vaccines based on DCs loaded with a pool of peptides derived from melanoma antigens + TBVAs were also observed to exhibit greater anti-tumor efficacy vs. DCs loaded with B16 tumor lysate (containing both melanoma antigens + TBVAs) in association with a superior tumor-infiltrating lymphocyte (TIL) content and CD8^+^ T cell responses against tumor cells and pericytes/VECs flow-sorted from progressor B16 lesions. Finally, we noted that combination vaccines that incorporate both the CKM and anti-PD-L1 antibody yielded the greatest therapeutic benefit in association with epitope spreading and the highest levels of multifunctional CD8^+^ TIL with reduced Treg/MDSC regulatory cell content in the tumor microenvironment (TME). These results support the translation of such combination vaccine strategies into the clinic for the treatment of patients with melanoma and other forms of solid cancer.

## 2. Materials and Methods

*Animals and cell lines*: Female C57BL/6J mice aged between 6 and 8 weeks were purchased from Jackson Laboratory (Bar Harbor, ME, USA). The B16.F10 (CRL-6475) murine melanoma and EL4 thymoma (TIB-39) cell lines were purchased from ATCC (Manassas, VA, USA) and cultured in Roswell Park Memorial Institute (RPMI)-1640 media (ThermoFisher Scientific, Waltham, MA, USA) supplemented with 10% heat-inactivated fetal bovine serum (Sigma-Aldrich, St. Louis, MO, USA), 100 µg/mL streptomycin, 100 U/mL penicillin (ThermoFisher Scientific), and 10 mmol/L L-glutamine (ThermoFisher Scientific) in a humidified incubator under 5% CO_2_ tension and 37 °C (i.e., complete RPMI media [CM]). All cell lines were free of mycoplasma contamination based on screening with a commercial Mycoplasma PCR Detection Kit (Abcam, Cambridge, MA, USA) per the manufacturer’s instructions.

*Peptides*: Peptides (and corresponding H-2 class I presenting alleles, AA sequences) included (i.) melanocyte/melanoma-associated peptides mgp100_25–33_ (D^b^; EGSRNQDWL) [19], mTRP1_455–463_ (D^b^; TAPDNLGYA) [20], mTRP2_180–188_ (K^b^; SVYDFFVWL) [21], (ii.) mutated B16 neoantigen-derived peptides included mutKIF18B/MUT30_735–744_ (D^b^; VDWENVSPEL), mutTNPO3/MUT17_503–510_ (K^b^; LAYLMKGL), and mutCPSF3l/MUT44_310–319_ (D^b^; RTFANNPGPM), as previously described [22], (iii.) TBVA-derived peptides mDLK1_262–270_ (K^b^; SGYGLTYRL; Appendix A), mEphA2_671–679_ (D^b^; FSHHNIIRL) [23], mHBB_34–42_ (K^b^; VVYPWTQRY; Appendix A), mNRP1_864–871_ (K^b^; MSALGVLL; Appendix A), mRGS5_71–78_ (K^b^; NSYGFASF; Appendix A), and mTEM1_76–85_ (D^b^; VGPANGLLWI; Appendix A), and (iv.) irrelevant control peptide HPV-16 E7_49–57_ (H-2D^b^; RAHYNIVTF) [24]. All peptides were synthesized by the University of Pittsburgh Peptide and Peptoid Synthesis Core Facility (a shared resource) and confirmed for a purity > 95% by the University of Pittsburgh Cancer Mass Spectrometry Core Facility (a shared resource).

*Tumor Lysate*: Tumor lysate was generated from cultured B16 melanoma cells using the freeze–thaw method as previously described [25]. Briefly, cells were resuspended in 2 mL of PBS and lysed by five sequential freeze (on methanol and dry ice)–thaw (room temperature) cycles, with cell disruption monitored by trypan blue staining. After further sonication of lysates for 10 min, specimens were centrifuged at 15,000 × *g* (30 min, 4 °C) and supernatants were recovered, quantitated for protein content by spectrophotometer (OD280_nm_), and stored at −80 °C until use.

*Generation of bone marrow-derived DCs and DC-based vaccines*: DCs were generated as previously described [9,10]. Briefly, bone marrow precursors isolated from the tibias and femurs of mice were cultured for 5 days in CM supplemented with 1000 U/mL rIL-4 and 1000 U/mL rGM-CSF (both from Peprotech, Cranbury, NJ, USA) to generate DCs. The resulting DCs were then purified using CD11c microbeads (STEMCELL Technologies, Cambridge, MA, USA) per the manufacturer’s instructions. To generate DC/adenovirus-based vaccines, purified DCs were infected with rAd.mIL-12p70 at a multiplicity of infection (MOI) = 50 and cultured for 48 h in rIL-4 and rGM-CSF-supplemented CM (yielding DC.IL12) as previously described [9,10]. While control DCs produced < 62.5 pg IL-12p70/mL/48 h/10^6^ cells, DC.IL12 cells produced 1–10 ng IL-12p70/mL/48 h/10^6^ cells [9,10]. To generate DC–peptide-based vaccines, DC.IL12 cells were pulsed with tumor lysate (300 μg) or pools of synthetic peptides (as indicated in text and figures with pools containing 1 μM of each peptide) 24 h after rAd.mIL-12p70 infection and left overnight at 37 °C. DC vaccine cells were harvested, washed with PBS, and injected (10^6^ DC in 50 μL PBS) s.c. in tumor-bearing mice as outlined below.

*Tumor Models*: Mice received subcutaneous (s.c.) injections of 10^5^ syngeneic B16 melanoma cells in 100 µL of PBS in the right flank. Ten days after inoculation, tumors were measured, and mice were randomized into cohorts of five–ten mice each with comparable mean tumor sizes. Mice were then left untreated, or they were treated with (i.) DC–peptide or DC–Lysate vaccines (10^6^ DC injected s.c. in the left flank contralateral to tumors), (ii.) CKM (Rintatolimod [Ampligen, 50 μg, i.p., from AIM Immunotech, Ocala, FL, USA] + rmIFNα [10,000 U, i.p., from R&D Systems, Minneapolis, MN, USA] + celecoxib/COX2i [75 μg, i.p., from Selleck Chemicals, Houston, TX, USA]), and/or (iii.) anti-PD-L1 antibody (100 μg, i.p., from BioXCell, Lebanon, NH, USA). Identical treatments were administered on days 17 and 24 post-tumor inoculation. Tumor growth was monitored daily and measured (two dimensions; long axis and short axis) every 3–4 days using a Vernier caliper and is reported as tumor area (in mm^2^ ± SD) based on the product of orthogonal measurements of the long and short axes of the palpable tumor. All mice were monitored, treated, and euthanized according to a University of Pittsburgh Institutional Animal Care and Use Committee approved protocol (# 24013014, approved 01/26/2024).

*T Cell Response Assays*: Spleens were harvested from mice 7 days after the second DC-based vaccination. Splenocytes were then stimulated in vitro with individual peptides (1 μM) for 48 h at 37 °C, 5% CO_2_, after which cell-free supernatants were analyzed for mIFNγ content using a cytokine-specific ELISA (BD Biosciences). To assess the specificity of TILs, CD8^+^ T cells (10^5^) freshly isolated from enzymatically digested day 24 B16 tumors were cocultured with (i.) flow-sorted tumor pericytes (10^4^) + VEC (10^4^) [designated as PVEC], (ii.) 2 × 10^4^ B16 melanoma cells, or (iii.) 2 × 10^4^ EL4 thymoma cells in U-bottom 96-well plates (Sigma-Aldrich) for 4–5 h at 37 °C. Harvested cells were then analyzed for intracellular cytokine staining by flow cytometry (IFN-γ, TNF-α, IL-2), as previously described [26].

*Tumor tissue processing for flow cytometry analysis*: Tumors were resected on day 24 post-tumor inoculation, minced, and digested using a cocktail of enzymes [RPMI containing DNAse I (20 U/mL), Collagenase IA (0.5 mg/mL), Collagenase II (0.5 mg/mL), and Collagenase IV (0.5 mg/mL); all from Sigma-Aldrich] for 30 min at 37 °C on a shaker. Tumor digests were then dissociated through a 70-micron filter (ThermoFisher). Purified cell populations and tumor digests were washed twice with PBS then blocked with FcR block (BD Pharmingen) prior to staining. Cells were then incubated with appropriate primary antibodies in FACS buffer for 30 min at 4 °C prior to fixation and flow cytometry analysis performed using either BD LSR II or BD Fortessa machines. Flow cytometry data were acquired using BD FACSDiva software and analyzed using FlowJo V.10.

*Isolation of PVEC*: To isolate tumor-associated pericytes and VECs (Appendix A), enzymatically digested day 17 tumor cell suspensions from untreated animals were stained with AlexaFluor 488-conjugated anti-mouse NG2 (Abcam), PE-conjugated anti-mouse CD31 (Biolegend, San Diego, CA), PerCP/Cy5.5-conjugated anti-mouse CD45 (Biolegend), and APC-conjugated anti-mouse CD140b/PDGFRβ (Biolegend). After washing with PBS, cells were sorted into populations enriched in pericytes (CD31^neg^CD45^neg^CD140b/PDGFRβ^+^NG2^+^) or VECs (CD31^+^CD45^neg^CD140b/PDGFRβ^neg^NG2^neg^) using a multicolor fluorescence-activated cell sorter (FACSAria; BD Biosciences). Our gating strategy is depicted in Appendix A. In all cases, cells were >95% pure for the stated phenotype, with qRT-PCR analyses documenting selective expression of the *DLK1*, *HBB*, *NG2*, *NRP1*, *RGS5*, and *TEM1* gene transcripts by isolated tumor pericytes, the *EphA2* and *CD31* gene transcripts by VECs, and the *gp100* gene transcript by B16 melanoma cells, with EL4 cells failing to express significant levels of any of these transcripts (Appendix A).

*TME Imaging by Immunofluorescence microscopy*: Tumor tissues were processed and stained as previously described [10]. Fluorescently labeled antibodies used to stain tumor tissue sections included the following: Alexa 488-conjugated rabbit anti-mouse CD8α (Abcam), Alexa 594-conjugated hamster anti-mouse CD11c (Biolegend), Alexa 488-conjugated rabbit anti-mouse CD31 (Abcam), and PE-conjugated rat anti-mouse CXCL10 (R&D Systems). Fluorescence images were acquired using a Nikon 90i microscope, with quantitation of fluorescent probes performed using Nikon Elements AR software. Post-acquisition, statistical analyses of fluorescent images were performed on GraphPad Prism V.10.

*Statistical considerations:* Comparisons between two groups were performed using two-tailed Student’s t-tests while comparisons between multiple groups were performed using (one-way or two-way) analysis of variance with Tukey’s post hoc analysis. Differences in survival curves were compared using the Mantel–Cox test. *p* values < 0.05 were considered significant. Prism V.10 (GraphPad Software, La Jolla, CA, USA) was used to generate graphs and to perform statistical tests.

## 3. Results

### 3.1. Coordinate Vaccine Targeting of Melanoma and Tumor-Associated Stromal Antigens Effectively Controls Melanoma Growth In Vivo

To address the comparative anti-tumor efficacy of a range of therapeutic vaccines, we first established B16 melanomas s.c. in the right flanks of C57BL/6 mice for 10 days and then treated animals with s.c. DC-based vaccines injected on the left (contralateral) flank on days 10, 17, and 24 (Figure 1A). To generate DC vaccines, syngenic BM precursor cells were cultured in media containing rmGM-CSF + rmIL-4 for 5 days before MACS isolation of CD11c^+^ DC. These DCs were then infected with a recombinant adenovirus encoding mIL-12p70 to develop DC.IL12 cells over a 48 h period. Finally, DC.IL12 cells were loaded with B16 lysates or synthetic H-2K^b^/D^b^-presented peptides derived from B16-expressed antigens (i.e., neoantigens [NeoAg; mutKIF18B, mutTNPO3, mutCPSF3l] or shared melanoma-associated antigens [MAA; mgp100, mTRP1, mTRP2]) and/or tumor-associated vascular pericyte/VEC antigens (TBVA; mDLK1, mEphA2, mHBB, mNRP1, mRGS5, mTEM1) as described in Materials and Methods. DC.IL12 cells were used as the preferred biologic adjuvant in our vaccine formulations based on our previous report validating their CD4^+^ T cell-independent ability to effectively (cross-)prime CD8^+^ T cell responses in vivo [9].

In our studies, we observed that all Ag-loaded DC-based vaccines effectively slowed the growth of well-established B16 tumors (Figure 1B) and extended the time to required euthanasia as a surrogate index of overall survival (Figure 1C) when compared to B16-bearing mice that were left untreated or that received s.c. injections of the control, Ag-unloaded DC.IL12 cells (*p* < 0.05). Amongst the DC-based vaccines evaluated, those incorporating peptides derived from tumor blood vessel antigens (TBVAs; pericyte or VECs) provided the greatest degree of therapeutic benefit (Figure 1B,C; *p* < 0.05 for DC-TBVA Peptide or DC–Pool Peptide vaccines vs. all other cohorts), with vaccines including both vascular and tumor antigen-derived peptides (i.e., NeoAg + MAA + TBVA pooled peptides) demonstrating somewhat greater anti-tumor efficacy when evaluating tumor growth at late time points (Figure 1B; *p* < 0.05 for DC–Pool Peptides vs. DC-TBVA Peptide vaccines on days > 38), but this did not translate into a significant extension in animal survival (Figure 1C; NS for DC–Pool Peptides vs. DC-TBVA Peptide vaccines).

To assess the correlation between vaccine anti-tumor efficacy and specific CD8^+^ T cell responses in vivo., melanomas were harvested from animals at day 24 and enzymatically digested into single cell suspensions. MACS-CD8^+^ TILs were quantitated per gram tumor tissue (Figure 1D) and portions of these effector cells were stimulated for 5 h in vitro with (i.) cultured B16 melanoma cells (Figure 1E) or (ii.) a 1:1 mixture of CD31^neg^CD45^neg^CD140b/PDGFRβ^+^NG2^+^ pericytes + CD31^+^CD45^neg^CD140b/PDGFRβ^neg^NG2^neg^ VECs that had been flow-sorted from day 17 B16 tumors established in the control, untreated C57BL/6 mice (Figure 1F and Appendix A) to assess the percentage of specific effector cells reactive against each target cohort based on IFN-γ intracellular cytokine staining (ICS) as monitored by flow cytometry. We observed that vaccines exhibiting the greatest degree of tumor growth control (Figure 1B) contained the highest levels of CD8^+^ TILs (Figure 1D), with all Ag-loaded vaccines promoting enhanced levels of CD8^+^ TILs (*p* < 0.05) when compared to tumors isolated from untreated mice or mice vaccinated with Ag-unloaded DC-based vaccines. The ICS results demonstrate the strongest anti-B16 tumor cell responses amongst CD8^+^ TILs recovered from animals receiving DC-MAA Peptide or DC–Pool (which include MAA) Peptide vaccines (Figure 1E), with all antigen-loaded DC-based vaccines eliciting statistically elevated T cell responses against B16 melanoma cells vs. untreated or Ag-unloaded DC vaccinated mice (*p* < 0.05). Predictably, vaccines containing TBVA Peptides (including the DC–Pool Peptide cohort) stimulated the strongest CD8^+^ TIL responses against tumor-derived pericytes or VECs that naturally express these antigens (Figure 1F; *p* < 0.0015 for TBVA Peptide-inclusive vaccines vs. all other treatments, with *p* = 0.44 for DC-TBVA Peptide vs. DC–Pool Peptide vaccines), although detectable pericyte/VEC-reactive CD8^+^ TILs were also identified in the DC–Lysate, DC-NeoAg Peptide, and DC-MAA Peptide vaccine cohorts. Notably, by comparing CD8^+^ TILs in the DC–Lysate and DC–Pool Peptide vaccines (which each contain both melanoma and pericyte/VEC antigens) for their reactivity against B16 tumor cells vs. tumor-derived pericytes/VECs, the resultant TIL repertoires appeared quantitatively balanced (in the case of DC–Lysate vaccines) or slightly skewed towards pericytes/VEC (in the case of DC–Pool Peptide vaccines) (Figure 1E,F). As a specificity control, CD8^+^ TILs isolated from all vaccine cohorts failed to produce/express IFN-γ in response to in vitro stimulation with irrelevant EL4 thymoma cells that fail to express any of the vaccine-targeted antigens (Appendix A).

Anti-tumor efficacy of DC-based vaccines coordinately targeting cancer cell- and tumor-associated pericyte/VEC antigens is improved by CKM and/or anti-PD-L1 blockade. Although our DC-based vaccines were able to extend the animal median survival from ~24 days (untreated) to a maximum of ~41 days (Figure 1C), we next chose to assess combination vaccine approaches to improve the benefits of these immunotherapies. For these experiments, we chose to focus on two different antigen formats that allowed for the coordinate targeting of both tumor cells and tumor-associated pericytes/VECs in vaccine formulations (i.e., DC–Lysate and DC–Pool Peptide vaccines). We then selected two interventions to combine with and potentially improve the anti-tumor efficacy of our vaccine formulations (Figure 2A): (i.) a chemokine-modulating (CKM) regimen consisting of i.p. delivered TLR3 agonist Ampligen (Rintatolimod; Poly I:Poly C12U), rmIFN-α, and Celecoxib (COX-2i) based on the ability of this treatment modality to promote a pro-inflammatory TME conducive for enhanced CD8^+^ T cell recruitment and coordinately reduced Treg/MDSC content [17] and/or (ii.) anti-PD-L1 to sustain anti-tumor CD8^+^ TIL (poly)functionality and prevent effector cell exhaustion [27,28].

### 3.2. Immunogenic and Immunomodulatory Activity of DC-Based Vaccines Combined with CKM and/or Anti-PD-L1 Blockade

Immunofluorescence microscopy (IFM) analyses of day 24 tumors isolated from B16-bearing mice treated with DC–Pool Peptide vaccines revealed elevated infiltration by CD8^+^ TILs (Figure 3A,B) and CD11c^+^ DC (Figure 3A,C) vs. tumors recovered from the control untreated mice, with further increased infiltration observed when vaccine treatments included administration of anti-PD-L1, CKM, or (particularly) anti-PD-L1 + CKM (Figure 3A–C). Consistent with our past experiences with TBVA-targeted vaccines that promote vascular normalization within the TME [9,10], therapeutic DC–Pool Peptide vaccination led to reduced CD31 blood vessel content in B16 tumors, which was further diminished in the case of combined vaccination with anti-PD-L1 and/or CKM (Figure 3D,E). It was noteworthy that treatment with CKM +/− anti-PD-L1, but not anti-PD-L1 alone also reduced CD31 vessel content in B16 tumors in the absence of vaccination (Figure 3D,E). Finally, in situ analysis of the CXCL10 content in the therapeutic TME using IFM revealed substantial upregulation in expression of this important CD8^+^ (CXCR3^+^) T cell-recruiting chemokine after treatment with all interventions evaluated except for anti-PD-L1 monotherapy (Figure 3D,F). Consistent with the CD8^+^ TIL content data (Figure 3B), the greatest levels of local CXCL10 production in the TME were observed in animals treated with the DC–Pool Peptide vaccine + anti-PD-L1 + CKM (*p* < 0.05 vs. all other cohorts; Figure 3F).

We next performed a more detailed analysis of the tumor immune microenvironment in B16-bearing mice treated with DC–Pool Peptide vaccines +/− anti-PD-L1 +/− CKM (Figure 4). MACS-CD8^+^ TILs were analyzed for polyfunctional reactivity against B16 tumor cells (Figure 4A), flow-sorted B16-associated NG2^+^CD31^neg^PDGFRB^+^CD45^neg^ pericytes + NG2^neg^CD31^+^PDGFRB^neg^CD45^neg^ VECs (PVEC; Figure 4B), or (antigenically) irrelevant EL4 thymoma cells (Appendix A). CD8^+^ TIL polyfunctionality was predicated on coordinate CD8^+^ TIL responder intracellular expression of IFNγ, TNFα, and IL2 as determined by flow cytometry after a 5 h in vitro stimulation with either tumor cells or PVECs. Although combining anti-PD-L1 with DC–Pool Peptide vaccines failed to significantly influence (*p* > 0.05) the levels of polyfunctional CD8^+^ TILs recognizing either B16 or PVEC target cells, vaccines combined with either CKM or CKM + anti-PD-L1 antibody resulted in elevated frequencies of polyfunctional CD8^+^ TILs reactive against B16 tumor cells or PVECs (Figure 4A,B) but not EL4 tumor cells (Appendix A).

Additional flow cytometric analyses revealed that (i.) DC–Pool Peptide vaccination significantly reduced the CD4^+^Foxp3^+^ Treg and CD11b^+^Gr-1^+^ MDSC content in the TME (Figure 4C,D), and (ii.) only the addition of CKM or CKM + anti-PD-L1 antibody to the DC–Pool Peptide vaccine regimen resulted in a further significant reduction in Treg and MDSC frequencies in B16 tumors when compared to treatment with DC–Pool Peptide vaccination alone.

## 4. Discussion

The major advances in the current study include the (i.) definition of novel (H-2^b^ class I-presented) epitopes derived from TBVAs including mDLK1, mHBB, mNRP1, mRGS5, and mTEM1 recognized by CD8^+^ T cells, (ii.) comparison of antigen classes/formats in vaccines targeting intrinsic tumor cell and/or tumor PVEC antigens for therapeutic efficacy, (iii.) screening of vaccine-induced CD8^+^ T cells for direct recognition of tumor cells or tumor PVECs in vitro, and (iv.) findings that vaccines coordinately targeting both tumor cells and tumor PVECs can be significantly improved by combining these treatment regimens with CKM or anti-PD-L1, but particularly with combined CKM + anti-PD-L1.

DC–Lysate/Peptide vaccines targeting intrinsic melanoma or PVEC antigens were effective in slowing tumor progression and extending overall survival in B16 models, with peptide-based vaccines outperforming lysate-based vaccines (as assessed) for these study endpoints. The reason for the superiority of peptide-based vaccines in the current studies could involve higher levels of MHC class I occupancy for specific peptide species achieved via the ex vivo loading of DCs using peptides vs. tumor lysates that require antigen-presenting cell uptake and subsequent processing. Alternatively, or additionally, tumor lysates may contain cancer-irrelevant peptides/proteins that compete for MHC-I loading on DCs or tumor-associated factors that limit effective antigen loading or the immunogenicity of vaccine DCs. Furthermore, this difference could reflect our empirical study design chosen for DC loading with tumor lysates that may not be fully optimized for DC-based vaccination to elicit anti-tumor T cell responses in vivo despite the dose of lysate used to load DCs in vaccine preparation being consistent with that used in other published studies using B16 melanoma models [29]. We plan to address these issues in future studies.

Therapeutic vaccine efficacy was associated with improved CD8^+^ TIL content, with these T cells producing IFN-γ in response to in vitro stimulation with tumor cells or flow-sorted populations of tumor-associated PVECs, but not model-irrelevant EL4 thymoma cells. As expected, DC–Lysate and DC–Pool Peptide vaccines (containing both tumor and PVEC antigens) were effective in eliciting/recruiting CD8^+^ TILs capable of recognizing both B16 melanoma cells and tumor-associated PVECs; however, this was also true to a variable degree for vaccines selectively targeting tumor- or PVEC-associated antigens. Hence, DC-NeoAg peptide and DC-MAA peptide vaccines prompted the development of detectable anti-PVEC TIL responses, and DC-TBVA peptide vaccines promoted significant levels of CD8^+^IFN-γ^+^ TILs reactive against B16 tumor cells, with no vaccines observed to promote T cells reactive against EL4 tumor cells. These data are consistent with an epitope spreading paradigm in which vaccine-induced T cells promote a pro-inflammatory TME that is conducive for local DC-mediated cross-priming of a more diversified T cell repertoire reactive against antigens that derive not only from tumor cells, but from the broad range of cell types that make up and are conditioned within the TME [13,30,31,32,33]. In this regard, a diversified therapeutic T cell repertoire that reacts against TME antigens beyond tumor-associated antigens conceptually and operationally affords host protection against cancer cells that represent antigen- or MHC-loss variants, which might otherwise evade T cell surveillance strictly directed against intrinsic tumor antigens [9,34].

Despite the ability of vaccines coordinately targeting both melanoma and PVEC antigens (i.e., DC–Lysate vs. DC–Pool Peptide vaccines) to provide therapeutic benefit, neither treatment modality resulted in tumor-free mice or to a survival extension beyond day 50 in our studies, necessitating the testing of combined immunotherapy approaches that potentiated a pro-inflammatory TME conducive to improved T cell infiltration (i.e., CKM regimen) and/or improved TIL maintenance/function (i.e., anti-PD-L1). We observed that the inclusion of CKM or CKM + anti-PD-L1 in a combined treatment regimen improved the anti-tumor efficacy of both DC–Lysate and DC–Pool Peptide vaccines, with the addition of anti-PD-L1 (in the absence of CKM) solely improving the treatment benefits significantly for the DC–Pool Peptide (but not DC–Lysate) vaccine cohort. Remarkably, inclusion of both CKM and anti-PD-L1 to either DC–Lysate or DC–Pool Peptide vaccines extended the median survival of treated animals beyond day 50. Although all tumor-bearing animals receiving combined DC–Lysate vaccines required euthanasia by day 75, 70% of mice in cohorts treated with DC–Pool Peptide vaccines + CKM + anti-PD-L1 were alive on study day 75 (vs. 20% and 0% alive in the DC–Pool Peptide + CKM only and DC–Pool Peptide + anti-PD-L1 only vaccine cohorts, respectively; both *p* < 0.05). Notably, we did not observe evidence for vaccine-induced autoimmune pathology (i.e., vitiligo, weight loss) in any animals treated in this study (data not shown).

In the setting of combination DC–Pool Peptide vaccines, improved treatment benefits in our B16 melanoma model were associated with a pro-inflammatory TME characterized by increased CD8^+^ TIL and CD11c^+^ DC content, increased TME expression of CXCL10 (a recruiting chemokine for activated CXCR3^+^ T cells [35]), and reduced CD31^+^ VEC, CD4^+^Foxp3^+^ Treg, and CD11b^+^Gr1^+^ MDSC content. Importantly, we observed that while provision of anti-PD-L1 modestly impacted the vaccine promotion of polyfunctional (IFN-γ, TNF-α, IL-2) CD8^+^ TIL reactivity against B16 melanoma or tumor-associated PVEC targets in vitro, combined treatment with vaccines + CKM or, more so, vaccines + CKM + anti-PD-L1 yielded superior frequencies of polyfunctional CD8^+^ TILs recognizing B16 (with *p* < 0.05 for vaccine + CKM + anti-PD-L1 vs. vaccine + CKM). These responses were melanoma TME-focused, since polyfunctional CD8^+^ TIL reactivity against EL4 thymoma target cells was not detected in any treatment cohort. Mechanism(s) underlying the loss of Treg and MDSC content in the therapeutic TME that will be investigated in prospective studies likely include treatment-associated vascular normalization (leading to reduced tumor hypoxia) and reduced local expression of chemokines known to preferentially recruit/retain these regulatory cell populations into/within the TME [10,36,37].

When taken in aggregate, our data support the translational use of combined vaccine approaches that integrate (i.) antigens/peptides expressed by tumor cells or additional stromal cell populations (i.e., tumor pericytes and VECs) and (ii.) co-treatment regimens that augment pro-inflammatory chemokine (i.e., CXCL10) production in the TME for enhanced (vaccine-induced or indirectly cross-primed) CD8^+^ T cell recruitment and sustained/augmented TIL polyfunctionality. While our studies focused on CKM and anti-PD-L1 as vaccine co-treatments, we would expect that our vaccine combined with alternate immune checkpoint inhibitors [38,39] or conditioning agents (i.e., vascular normalizing agents, STING agonists, regulatory cell antagonists, amongst others) [40,41,42,43] that promote inflammation/IFN production in the TME would also yield enhanced anti-tumor efficacy when compared to component monotherapies. It is also important to note that sustained intervention benefits associated with the maintenance of TIL polyfunctionality/fate would be anticipated for treatment schedules that extend beyond day 24 (a limiting variable in the current studies). We plan to pursue such examinations as well as extended treatment regimens in future.

It is important to acknowledge the presence of additional tumor stromal cell populations that are aberrantly conditioned in the TME, including adipocytes (CAAs) [44], fibroblasts (CAFs) [45,46,47], and mesenchymal stem cells (MSCs) [48,49]. It is entirely possible that vaccines coordinately targeting antigens differentially expressed by these cell populations could strengthen the immunogenicity and anti-tumor efficacy of combination vaccines that also include tumor-associated and tumor PVEC-associated peptides/antigens. At a minimum, T cell responses against such stromal cell-associated antigens could be monitored as indices for clinically relevant epitope spreading on treatment with a range of immunotherapies. Finally, as the list of vaccine adjuvants and conditioning regimens reported to improve TIL recruitment and/or to reduce TIL exhaustion is ever-expanding, prospective studies to define optimally effective (and safe) therapeutic combination vaccine approaches are expected to remain a fertile area of translational cancer immunology research for the extended future.

## Figures and Tables

**Figure 1 vaccines-12-00777-f001:**
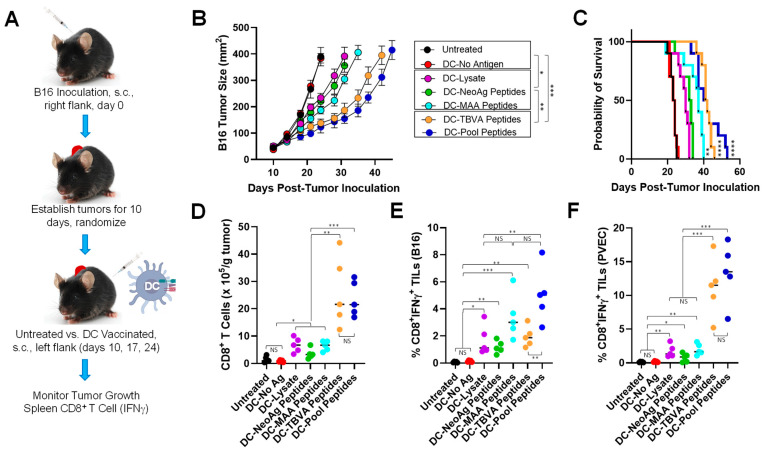
Therapeutic DC-based vaccination slows B16 melanoma growth and extends survival in association with increased frequencies of anti-tumor and/or anti-PVEC CD8^+^ TIL. (**A**) Vaccination schema and endpoint analyses are indicated with additional experimental details provided in Materials and Methods. (**B**) Tumor growth over time on treatment with the various DC-based vaccines (*n* = 10 mice/group). * *p* < 0.05, ** *p* < 0.01, and *** *p* < 0.001 (ANOVA). (**C**) Kaplan–Meier plots of time to required euthanasia as a survival index. ** *p* < 0.01 and **** *p* < 0.0001 (Mantel–Cox) vs. DC–Lysate (*n* = 10 mice/group). (**D**) Numbers of CD8^+^ T cells in harvested tumors (d24) were determined after enzymatic tissue digestion and performance of flow cytometry as described in Materials and Methods. In (**E**,**F**), MACS-CD8^+^ TILs were analyzed for reactivity against B16 tumor cells or a 1:1 mixture of pericytes and VECs previously flow-sorted from enzymatically digested (untreated) B16 tumors by intracellular cytokine (IFNγ) staining monitored by flow cytometry, respectively. * *p* < 0.05, ** *p* < 0.01, *** *p* < 0.001 (Student’s *t* test).

**Figure 2 vaccines-12-00777-f002:**
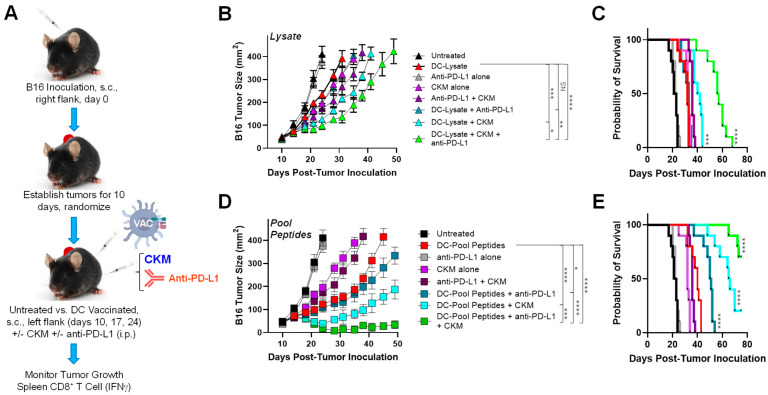
The anti-tumor efficacy of vaccines coordinately targeting both intrinsic melanoma and PVEC antigens is improved by co-administration of CKM and/or anti-PD-L1 antibody. (**A**) Treatment schema and endpoint analyses are indicated with additional experimental details provided in Materials and Methods. B16 tumor 3-based vaccines to control B16 melanoma growth (**B**) and extend survival (**C**) in vivo was not significantly improved by co-treatment with single agent anti-PD-L1; however, vaccination with DC–Lysate + CKM or DC–Lysate + anti-PD-L1 + CKM yielded improved treatment outcomes. The combination vaccine including both anti-PD-L1 and CKM yielded the greatest degree of therapeutic benefit. Similarly, the anti-melanoma efficacy of DC–Pool Peptide vaccines was enhanced by co-treatment with CKM and more so by CKM + anti-PD-L1 (**D**,**E**). Combination vaccines including anti-PD-L1 without CKM provided a significant, but only modest benefit over DC–Pool Peptide vaccination alone at late time points. When comparing combination DC–Lysate vaccines to DC–Pool Peptide vaccines that incorporate both CKM and anti-PD-L1, it was noteworthy that only the peptide-based vaccines resulted in surviving animals on day 75 (i.e., 70% survival for DC–Pool Peptide vs. 0% DC–Lysate vaccines, (**C**,**E**)). * *p* < 005, ** *p* < 0.01, *** *p* < 0.001, and **** *p* < 0.0001.

**Figure 3 vaccines-12-00777-f003:**
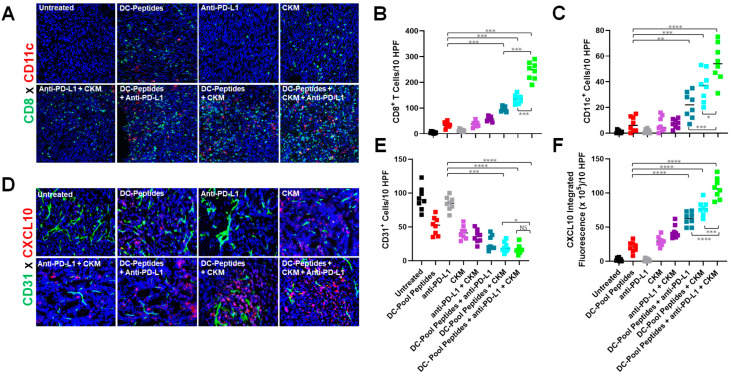
Therapeutic DC–Pool Peptide vaccines promote reduced vascular content and coordinate increased levels of CD8^+^ T cells and CD11c^+^ DC and increased expression of CXCL10 in the B16 TME. Day 24 tumors harvested from DC–Pool Peptide vaccine cohorts described in Figure 2 were analyzed by immunofluorescence microscopy for their content of CD8^+^ TIL (**A,B**), CD11c^+^ DC (**A,C**), CD31^+^ vessels (**D,E**), and CXCL10 (**D,F**). Experimental details are provided in Materials and Methods (*n* = 8 mice/group). * *p* < 005, ** *p* < 0.01, *** *p* < 0.001, and **** *p* < 0.0001 (Student’s *t* test). (**E**,**F**).

**Figure 4 vaccines-12-00777-f004:**
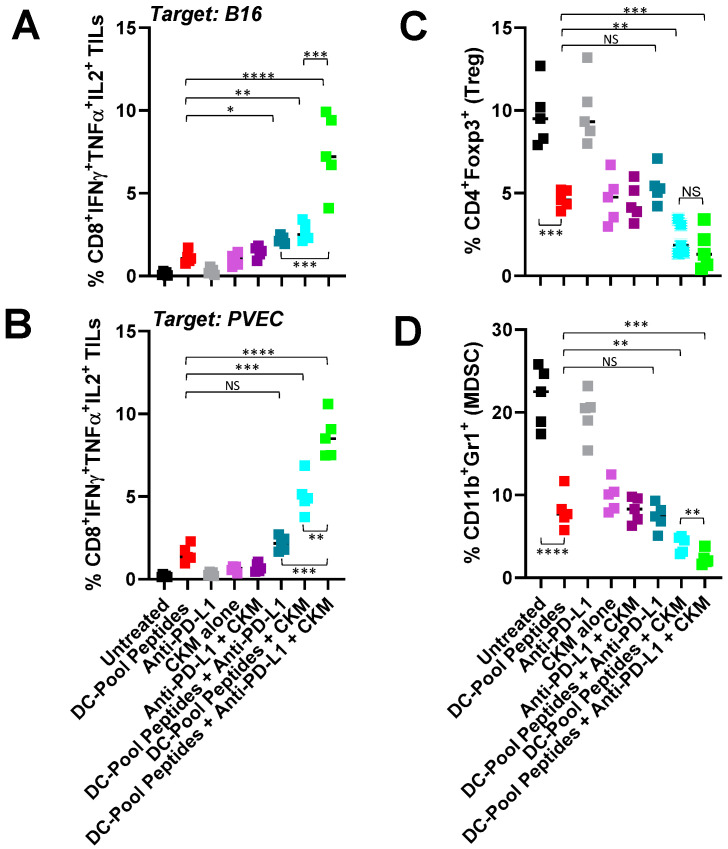
B16 tumors in mice treated with therapeutic DC–Pool Peptide vaccines exhibit CD8^+^ T cells with improved polyfunctional responses against tumor cells and PVECs and a coordinate reduction in Treg and MDSC content. Day 24 tumors were harvested from DC–Pool Peptide vaccine cohorts described in Figure 2, enzymatically digested and single cell suspensions recovered for analysis. MACS-CD8^+^ TIL isolated from these cell suspensions were analyzed for reactivity against B16 tumor cells (**A**) or PVECs (**B**) as outlined in Figure 1E, with the addition of intracellular cytokine staining for IFNγ, TNFα, and IL2. In (**C**) and (**D**), frequencies of CD4^+^Foxp3^+^ Treg and CD11b^+^Gr1^+^ MDSC were determined by flow cytometry from CD45^+^ gated tumor digest single cell suspensions. *n* = 5 mice/group. * *p* < 005, ** *p* < 0.01, *** *p* < 0.001, and **** *p* < 0.0001 (Student *t* test).

## Data Availability

The original contributions presented in the study are included in the article/supplementary material, further inquiries can be directed to the corresponding author.

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
