# Peer review of "Therapeutic Anti-Tumor Efficacy of DC-Based Vaccines Targeting TME-Associated Antigens Is Improved When Combined with a Chemokine-Modulating Regimen and/or Anti-PD-L1"

_vaccines, 2024, doi:10.3390/vaccines12070777_

Round 1

Reviewer 1 Report

Comments and Suggestions for Authors

The manuscript reveals the efficacy of DC-based vaccines targeting TME-associated antigens. The rationale is adequate. The objectives were clear, and the experiments were performed accordingly. The results are interesting, well-presented, and discussed.  The only issue that would probably be interesting to include is the limitation of maintaining the T-cell response in solid tumors. 

Author Response

We thank the reviewer for their positive assessment. Per their comment "The only issue that would probably be interesting to include is the limitation of maintaining the T-cell response in solid tumors". Response: On lines 451-453 of the revised Discussion we now state "It is also important to note that sustained intervention benefits associated with maintenance of TIL polyfunctionality/fate would be anticipated for treatment schedules that extend beyond day 24 (a limiting variable in the current studies)." Please see attachment.

Reviewer 2 Report

Comments and Suggestions for Authors

In the submitted manuscript by J. L. Taylor et al., the authors examined whether the therapeutic benefits of such tumor blood vessel antigen (TBVA)-targeted vaccines could be improved by co-targeting of tumor antigens in the s.c. B16 melanoma model. They found that DC-peptide or DC-tumor lysate vaccines coordinately targeting melanoma antigens and TBVA were effective in slowing B16 growth in vivo and extending survival, with superior outcomes observed for DC-peptide based vaccines. Peptide-based vaccines that selectively target either melanoma antigens or TBVA elicited a CD8+ T cell repertoire recognizing both tumor cells and tumor-associated VEC and pericytes in vitro, consistent with a treatment-induced epitope spreading mechanism. 

This is an interesting and well written study. However, some critical points should be addressed.

1. Fig. 4: It is not clear how the frequency of Treg and MDSC were determined (out of which parental population)?

2. Which mechanisms could be involved in the downregulation of Treg and MDSC frequencies in the TME? Please include these possible mechanisms in Discussion. 

Author Response

We thank the reviewer for their favorable assessment of this work. Per the 2 comments cited by Rev. #2 (and our response--please also see attached file):

  1. Fig. 4: It is not clear how the frequency of Treg and MDSC were determined (out of which parental population)? Response: These were determined from the gated CD45+ population of leukocytes in the tumor digests. This information is now indicated in the revised Fig. 4 legend.
  2. Which mechanisms could be involved in the downregulation of Treg and MDSC frequencies in the TME? Please include these possible mechanisms in Discussion. Response: We have now added the following text in our revised Discussion in lines 436-440 "Mechanism(s) underlying loss of Treg and MDSC content in the therapeutic TME that will be investigated in prospective studies likely include treatment-associated vascular normalization (leading to reduced tumor hypoxia) and reduced local expression of chemokines known to preferentially recruit/retain these regulatory cell populations into/within the TME10,36,37. "  

Reviewer 3 Report

Comments and Suggestions for Authors

This is a very good paper with robust data presented throughout.

 In Methods the authors indicate five mice were included into each treatment group however from the survival curves it is clear the groups contained more mice. Please correct and indicate in the legends how many mice were used in each TGI and survival experiment.

The authors indicated in the Methods that they used two types of image quantification software. Please provide more details which data was generated using one or the other and how this might affect the comparisons. Please also specify which statistical test was used in each experiment; this is not clear from the legends.  

Author Response

We thank the reviewer for their favorable assessment of this work. In response to Rev. #3's comments (please also see the attached file):

1.) In Methods the authors indicate five mice were included into each treatment group however from the survival curves it is clear the groups contained more mice. Please correct and indicate in the legends how many mice were used in each TGI and survival experiment. Response: 5-10 mice/group were analyzed in individual experiments. These are now specified in the Figure legends.

2.) The authors indicated in the Methods that they used two types of image quantification software. Please provide more details which data was generated using one or the other and how this might affect the comparisons. Response: Thank you for calling this to our attention. While 2 microscopes and 2 sets of software were used at various times in developing this study, all data reported in this manuscript were generated using the Nikon scope and Nikon software. This is now corrected in the revised Materials and Methods section of the manuscript.

3.) Please also specify which statistical test was used in each experiment; this is not clear from the legends.  Response: This information has now been added to all Figure legends as requested.